# Risk of acute myocardial infarction among new users of chondroitin sulfate: A nested case-control study

Ramón Mazzucchelli[1], Sara Rodríguez-Martín[2,3], Alberto García-Vadillo[4], Miguel Gil[5], Antonio Rodríguez-Miguel[2,3], Diana Barreira-Hernández[2,3], Alberto García-Lledó[6,7], Francisco J. de Abajo[2,3]*

1 Rheumatology Unit, Hospital Universitario Fundación Alcorcón, Alcorcón, Madrid, Spain, 2 Clinical Pharmacology Unit, University Hospital Príncipe de Asturias, Alcalá de Henares, Madrid, Spain, 3 Department of Biomedical Sciences (Pharmacology), School of Medicine and Health Sciences, University of Alcalá (IRYCIS), Alcalá de Henares, Madrid, Spain, 4 Rheumatology Department, Hospital Universitario La Princesa, Madrid, Spain, 5 Division of Pharmacoepidemiology and Pharmacovigilance, Spanish Agency of Medicines and Medical Devices (AEMPS), Madrid, Spain, 6 Department of Cardiology, University Hospital Príncipe de Asturias, Alcalá de Henares, Madrid, Spain, 7 Department of Medicine, University of Alcalá, Alcalá de Henares, Madrid, Spain

* francisco.abajo@uah.es

**Data Availability Statement:** All relevant data are within the paper and its Supporting Information files.

## Abstract

### Objective

To test the hypothesis that the use of chondroitin sulfate (CS) or glucosamine reduces the risk of acute myocardial infarction (AMI).

### Design

Case-control study nested in a primary cohort of patients aged 40 to 99 years, using the database BIFAP during the 2002–2015 study period. From this cohort, we identified incident cases of AMI and randomly selected five controls per case, matched by exact age, gender, and index date. Adjusted odds ratios (AOR) and 95% confidence interval (CI) were computed through a conditional logistic regression. Only new users of CS or glucosamine were considered.

### Results

A total of 23,585 incident cases of AMI and 117,405 controls were included. Of them, 89 cases (0.38%) and 757 controls (0.64%) were current users of CS at index date, yielding an AOR of 0.57 (95%CI: 0.46–0.72). The reduced risk among current users was observed in both short-term (<365 days, AOR = 0.58; 95%CI: 0.45–0.75) and long-term users (>364 days AOR = 0.56; 95%CI:0.36–0.87), in both sexes (men, AOR = 0.52; 95%CI:0.38–0.70; women, AOR = 0.65; 95%CI:0.46–0.91), in individuals over or under 70 years of age (AOR = 0.54; 95%CI:0.38–0.77, and AOR = 0.61; 95%CI:0.45–0.82, respectively) and in individuals at intermediate (AOR = 0.65; 95%CI:0.48–0.91) and high cardiovascular risk (AOR = 0.48; 95%CI:0.27–0.83), but not in those at low risk (AOR = 1.11; 95%CI:0.48–

**Funding:** This study was supported by a research grant from Instituto de Salud Carlos III—Ministerio de Ciencia e Innovación (# PI16/01353), granted to F.d.A., cofounded by FEDER.

**Competing interests:** FdA has received unrestricted research grants from Instituto de Salud Carlos III, Instituto Teófilo Hernando, Chiesi and Sanofi-Pasteur for other projects in the last 5 years. The rest of authors have no conflicts of interest to declare. The results, discussion, and conclusions are from the authors and do not necessarily represent the position of the Spanish Agency for Medicines and Medical Devices. This does not alter our adherence to PLOS ONE policies on sharing data and materials.

2.56). In contrast, the current use of glucosamine was not associated with either increased or decreased risk of AMI (AOR = 0.86; 95%CI:0.66–1.08).

## Conclusions

Our results support a cardioprotective effect of CS, while glucosamine seems to be neutral. The protection was remarkable among subgroups at high cardiovascular risk.

## 1. Introduction

Osteoarthritis (OA) and cardiovascular (CV) diseases are epidemiologically associated. In 2008, Hochberg [1], in a systematic review, reported a higher mortality risk in patients with OA as compared to the general population and suggested that it could be the result of a low-grade systemic inflammation, lack of physical activity, or both. These results were confirmed by Hawker *et al* [2] in a cohort of patients with symptomatic knee and/or hip OA and Barbour *et al*. [3] analyzing the association between hip radiographic OA and mortality.

Atherosclerosis is a chronic inflammatory disease characterized by activation of the immune system [4–8]. Throughout the evolution of the process, and due to endothelium inflammation, monocytes migrate from the bloodstream, infiltrate in atherosclerotic lesions, differentiate into macrophages and foam cells [9,10]. These cells produce proinflammatory mediators such as TNF-α and interleukin 1ß, which play a key role in the development and exacerbation of atherosclerosis [11,12]. Such inflammatory hypothesis has gained a strong support after the recent publication of two clinical trials [13,14].

Chondroitin Sulfate (CS) and Glucosamine (Sulfate or Hydrochloride), extensively prescribed in some countries, are classified among SYSADOAs (Symptomatic Slow-Acting Drugs for Osteoarthritis) a heterogeneous group of drugs reportedly to modify OA symptoms slowly and independently of nonsteroidal anti-inflammatory drugs (NSAIDs), analgesics or any other therapeutic option. SYSADOAs have always had zealous supporters and opponents for a variety of reasons [15,16]. In addition to CS and glucosamine this group includes hyaluronic acid (AH) and diacerein but both have little in common with CS and glucosamine. Also, CS and glucosamine are chemically different natural compounds (glucosamine is an amino sugar and chondroitin sulfate is a glycosaminoglycan) involved in proteoglycan biosynthesis [17,18]. Epidemiological studies have suggested that CS and glucosamine could play a role in cardiovascular disease (CVD) prevention [19–22], as well as reduction of mortality [21–23], colorectal cancer [21,24–26] and other diseases [19,21,27,28]. Studies to date have included prevalent users, therefore a bias that overestimates protection cannot be excluded. A way to avoid this bias is to only include patients who initiate treatment (new users) [29]. On the other hand, two randomized clinical trials carried out in the 70s observed that CS reduces early coronary events and late mortality [30–32].

The aim of this study was to test the hypothesis that the use of CS and glucosamine shows a protective effect against acute myocardial infarction (AMI) in a real-world setting.

## 2. Patients and methods

### 2.1. Data source and study design

We performed a case-control study nested in a primary cohort obtained from BIFAP (Base de datos para la Investigación Farmacoepidemiológica en Atención Primaria), a primary health

care database from Spain [33]. BIFAP contains anonymised electronic records on clinical events, prescriptions and laboratory tests, among others, that are recorded routinely by primary care practitioners. BIFAP-2016, which was used in this study, contains data from 7.6 million of patients (38.6 million subjects per year) with an average follow-up of 5.1 years, from nine different Spanish autonomous communities out of a total of 17. BIFAP reflects the distribution of the Spanish population by sex and age, and has been validated through numerous pharmacoepidemiological studies [20,34], therefore obtaining results that are comparable to other known European databases. The study period covers 14 years (from January 1st, 2002, to December 31st, 2015. In a first step, we constructed a primary cohort composed of all the patients registered in the database that they were aged 40 to 99 years, had at least 1 year of follow-up with their primary care physician and did not have a history of cancer or AMI. For all patients included in the primary cohort (n = 3,764,470), the first day they fulfilled all the inclusion criteria was considered as the "start date". Since then, follow-up was carried out until the occurrence of any of the following events: incident AMI, turning 100 years old, cancer diagnosis, death, or end of study period.

## 2.2. Selection of cases and controls

Among patients in the primary cohort, incident AMI cases were initially searched through codes and texts in diagnosis fields. As for disease classification, eight autonomous communities used the International Classification of Primary Care, Second Edition (ICPC-2), and one autonomous community used the International Classification of Diseases, Ninth Revision, Clinical Modification (ICD-9-CM). The search was done to identify all potential AMI cases in the primary care cohort which were defined by code ICPC-2 K75 (acute myocardial infarction), code ICD-9-CM 410.9 (myocardial infarction) or related terms (free text) in diagnostic field. Next, potential identified cases were clustered into homogenous subgroups according to available information and a random sample was extracted for each subgroup, manually validating a total of 600 cases. The validation consisted of looking for additional information in the patient's medical record (such as comments in the free text associated with the diagnosis, records of hospitalization, interventions performed on the patient such as angioplasty or thrombolysis, biochemical test results, electrocardiogram results,...) that would confirm that it was an incident case of AMI. Validation was performed independently by two of the researchers who were blind to any drug exposure and discrepancies were resolved by consensus of the whole research team. For each sample of validated cases in the homogeneous subgroups, the positive predictive value (PPV) was calculated, and when this was greater than 80%, the entire subgroup was considered valid cases. Overall, the PPV of the study was 87.2% (95% CI: 84.1–89.8%). The date when the first record of AMI took place was designated as "index date". Five controls per case were randomly selected from the underlying cohort following a risk-set sampling in which controls were individually matched with cases on exact age, sex and index date.

## 2.3. New user design

The analysis was performed among patients who initiated CS and glucosamine prescriptions (new users). This involved the exclusion of all cases and controls with recorded prescriptions of CS and/or glucosamine before the start date [29] (Fig 1).

## 2.4. Definition of exposure

SYSADOAs included in the present study are glucosamine (sulfate or hydrochloride) and CS. Doses considered were less than 800 mg/day or more for CS and 1,500 mg/day for

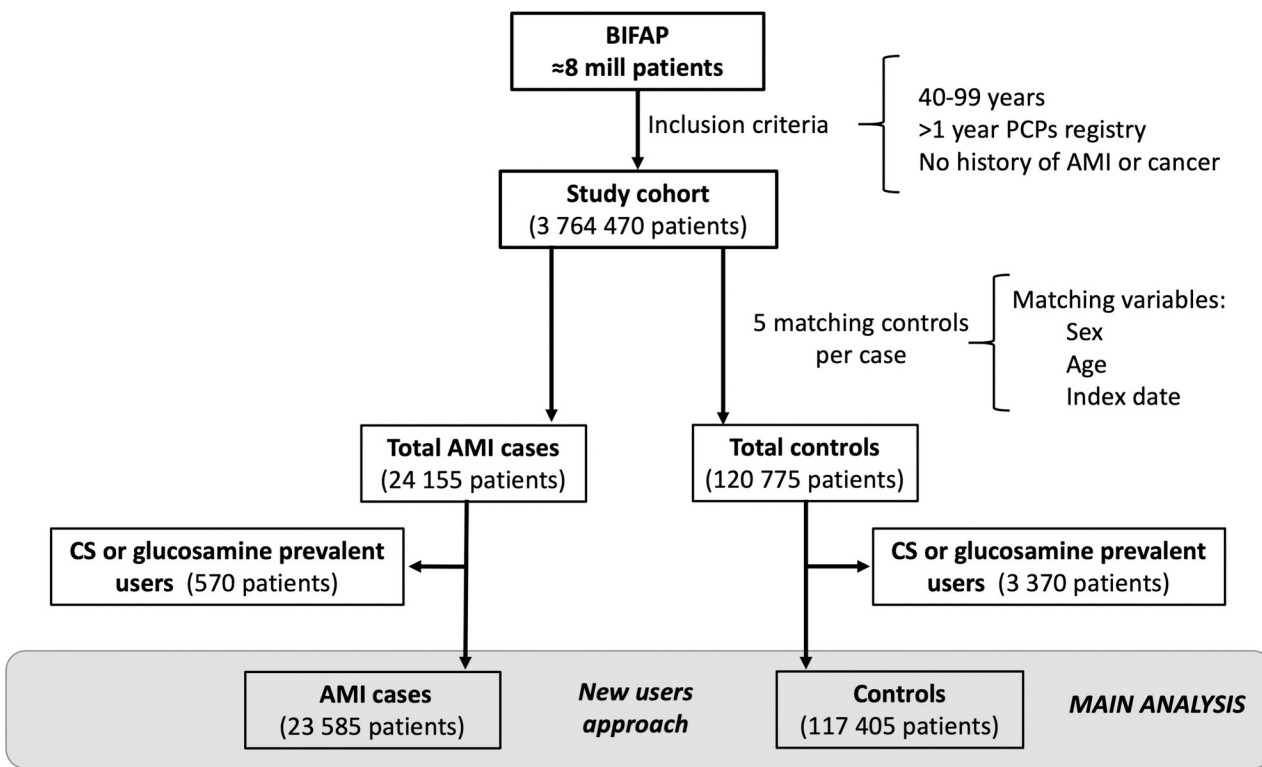

**Fig 1. Flowchart of patient selection.** Abbreviations: BIFAP:" Base de datos para la Investigación Farmacoepidemiológica en Atención Primaria";
AMI: Acute Myocardial Infarction; CS: Chondroitin Sulfate.

glucosamine. Patients were classified as "current users" of CS or glucosamine when the last
prescription finished within 30 days before the index date; "recent users" when it finished
between 31 and 365 days before the index date; "past users" when it finished more than 365
days before the index date; and "non-users" when there was no recorded prescription either
for CS or glucosamine before the index date.

Treatment duration was calculated for current users by adding consecutive prescriptions
(when the gap between the end of a prescription and the beginning of the following one was
no longer than 90 days). Afterwards, patients were clustered into two groups: less than 365
days, and 365 days or more (shorter durations were also explored).

## 2.5. Confounding factors

The selection of potential confounding variables was driven by expert knowledge, avoiding
data-driven methods. We ascertained the history of the following comorbidities and risk factors
any time before the index date: cerebrovascular disease (ischemic, hemorrhagic stroke or non-
specified and transient ischemic attack), heart failure, angina pectoris (recorded as such and/or
use of nitrates), peripheral arterial disease (PAD), hypertension, diabetes (recorded as such and/
or use of glucose-lowering drugs), dyslipidemia (recorded as such and/or use of lipid-lowering
drugs), rheumatoid arthritis, asymptomatic hyperuricemia or gout and chronic kidney disease.
Furthermore, the following factors were also considered: number of visits to primary care physi-
cian (PCP) during the year prior to index date; body mass index (BMI); smoking (current, past,
never and unknown); and current treatment with: antiplatelet drugs, oral anticoagulants, non-
steroidal anti-inflammatory drugs (NSAIDs), metamizole, paracetamol, calcium and vitamin D

supplements, proton pump inhibitors (PPIs), $H_2$-receptor antagonists, corticosteroids, angiotensin-converting enzyme inhibitors (ACEI), angiotensin II receptor blockers (ARB), calcium channel blockers (CCB), beta-blockers, alpha-blockers, and diuretics.

## 2.6. Statistical analysis

The association between the exposure to CS and glucosamine with incident AMI was evaluated by calculating the odds ratios (OR) and their 95% confidence intervals (95%CI) through a conditional logistic regression model. First, we estimated the non-adjusted OR by only including in the model the exposure of interest and the matching variables (age, sex and calendar year). Then, we estimated the fully adjusted odd ratios (AOR) including in the model all potential confounding factors mentioned above. Additionally, interaction with age (stratified as under 70, and 70 years or more), sex, concomitant use of NSAIDs and background CV risk were examined. The latter was defined as: *high risk* for patients with history of PAD, angina pectoris, cerebrovascular accident or diabetes; *intermediate risk* for patients with history of hypertension, dyslipidemia, chronic renal failure, current smoking or BMI>30kg/m$^2$ when they were not included in the high-risk group; and *low risk* for the rest. Patients with diabetes mellitus were included in the high risk group because it has been reported that its level of risk is equivalent to that of ischemic heart disease [35]. For statistical evaluation of interaction, we run fully adjusted models within different interacting variable categories, and AORs associated with the current use of drugs of interest in comparison with the non-use for each stratum were estimated. AORs of the different strata of the interacting variable were compared using the test of interaction described by Altman and Bland [36].

Additionally, in a secondary analysis, we compared current users of CS with current users of glucosamine.

We only reported the AORs when there were at least 5 exposed cases. Results were considered statistically significant when the p-value was lower than 0.05.

Missing values for specific variables such as smoking (45–50%) and BMI (36–39%) were addressed using a multiple imputation with chained equations model (MICE) [37] (see S1 Appendix in S1 File).

An analysis of potential collinearity was performed by measuring the variance inflation factor (VIF); according to this, collinearity is considered to be present when a variable has an independent VIF value above 10 or the mean VIF is above 6.

Analyses were performed with STATA version 15/SE software (StataCorp. College Station, TX, US).

## 2.7. Sensitivity analysis

A sensitivity analysis was also performed including prevalent users of CS and glucosamine.

## 2.8. Ethical aspects

The BIFAP Scientific Committee granted access to fully anonymised electronic medical records (project #04/2016; approval date May 26th, 2016). Afterwards, on July 1st, 2020, this committee approved specifically the analysis proposed for this study. Additionally, the Research Ethics Committee of the Hospital Fundación Alcorcon (Ref 20/76) approved the study on May 4th, 2020.

## 3. Results

A total of 23,585 incident cases of AMI and 117,405 matched controls were included (Fig 1). Characteristics are shown in Table 1. As expected, the prevalence of CV risk factors and the use of CV drugs was higher for cases when compared to controls.

**Table 1. Cases and controls characteristics.** SYSADOA.

| | Cases (%) N = 23585 | Controls (%) N = 117405 | Non-adjusted OR[‖] (95% CI) |
|---|---|---|---|
| Age; mean (SD) | 67.0 (13.4) | 66.9 (13.5) | - |
| Men | 16922 (71.75) | 84325 (71.82) | - |
| *Visits (last 12 months)* | | | |
| Up to 5 | 6806 (28.86) | 44370 (37.79) | 1 (Ref.) |
| 6–15 | 8795 (37.29) | 42317 (36.04) | 1.44 (1.39–1.50) |
| 16–24 | 4335 (18.38) | 17638 (15.02) | 1.81 (1.73–1.89) |
| 25+ | 3649 (15.47) | 13080 (11.14) | 2.14 (2.03–2.25) |
| *BMI kg/m²* | | | |
| Up to 24.9 | 2668 (11.31) | 14220 (12.11) | 1 (Ref.) |
| 25–29 | 6819 (28.91) | 33042 (28.14) | 1.10 (1.05–1.16) |
| 30–34 | 4026 (17.07) | 18068 (15.39) | 1.19 (1.13–1.26) |
| 35–39 | 1081 (4.58) | 4304 (3.67) | 1.35 (1.24–1.46) |
| 40+ | 327 (1.39) | 1099 (0.94) | 1.58 (1.39–1.81) |
| Unknown | 8664 (36.74) | 46672 (39.75) | 0.98 (0.94–1.03) |
| *Smoking* | | | |
| Never smoking | 5294 (22.45) | 30882 (26.30) | 1 (Ref.) |
| Current smoker | 6382 (27.06) | 19766 (16.84) | 2.03 (1.94–2.12) |
| Past smoker | 1265 (5.36) | 6933 (5.91) | 1.11 (1.04–1.19) |
| Unknown | 10644 (45.13) | 59824 (50.96) | 1.07 (1.03–1.11) |
| *CVA* | | | |
| Ischemic | 586 (2.48) | 2160 (1.84) | 1.38 (1.25–1.51) |
| Hemorrhagic | 87 (0.37) | 351 (0.30) | 1.26 (1.00–1.60) |
| Unspecified | 423 (1.79) | 1782 (1.52) | 1.21 (1.08–1.34) |
| TIA | 483 (2.05) | 1940 (1.65) | 1.27 (1.15–1.41) |
| Heart failure | 880 (3.73) | 3049 (2.60) | 1.48 (1.37–1.60) |
| Angina pectoris[*] | 2657 (11.27) | 5106 (4.35) | 2.90 (2.76–3.05) |
| PAD | 1075 (4.56) | 2424 (2.06) | 2.31 (2.15–2.49) |
| Hypertension | 12192 (51.69) | 50657 (43.15) | 1.49 (1.45–1.54) |
| Diabetes [†] | 6396 (27.12) | 19382 (16.51) | 1.92 (1.86–1.99) |
| Dyslipidemia [§] | 11.013 (46.69) | 41228 (35.12) | 1.67 (1.62–1.72) |
| Rheumatoid arthritis | 229 (0.97) | 730 (0.62) | 1.57 (1.35–1.82) |
| Chronic kidney failure | 893 (3.79) | 2836 (2.42) | 1.61 (1.49–1.74) |
| Hyperuricaemia (asymptomatic) | 4371 (18.53) | 17479 (14.89) | 1.32 (1.27–1.37) |
| Gout | 1135 (4.81) | 5023 (4.28) | 1.20 (1.12–1.28) |
| *Current use of* | | | |
| Antiplatelet drugs | 4651 (19.72) | 14198 (12.09) | 2.04 (1.96–2.12) |
| Oral anticoagulants | 900 (3.82) | 4881 (4.16) | 0.92 (0.85–0.99) |
| Paracetamol | 3076 (13.04) | 14038 (11,96) | 1.19 (1.13–1.24) |
| Metamizole | 934 (3.96) | 3325 (2.83) | 1.52 (1.41–1.64) |
| NSAIDs | 2327 (9.87) | 10454 (8.90) | 1.20 (1.14–1.27) |
| Calcium suppl (w/, w/o vit D) | 720 (3.05) | 3985 (3.39) | 0.89 (0.82–0.97) |
| Corticosteroids | 511 (2.17) | 1719 (1.46) | 1.52 (1.37–1.68) |
| ACE inhibitors | 4109 (17.42) | 16837 (14.34) | 1.37 (1.32–1.43) |
| ARBs | 3640 (15.43) | 14085 (12.00) | 1.42 (1.37–1.48) |
| CCBs | 3227 (13.68) | 11116 (9.47) | 1.63 (1.56–1.70) |
| Beta-Blockers | 2577 (10.93) | 7406 (6.31) | 1.91 (1.82–2.00) |
| Alfa-Blockers | 590 (2.50) | 2424 (2.06) | 1.22 (1.12–1.34) |

*(Continued)*

**Table 1.** (Continued)

| | Cases (%) N = 23585 | Controls (%) N = 117405 | Non-adjusted OR[‖] (95% CI) |
|---|---|---|---|
| Diuretics | 3016 (12.79) | 12152 (10.35) | 1.38 (1.32–1.44) |
| PPIs | 6235 (26.44) | 24317 (20.71) | 1.54 (1.48–1.59) |
| H$_2$ receptor blockers | 508 (2.15) | 1608 (1.37) | 1.62 (1.47–1.78) |

Abbreviations: ACE: Angiotensin Converting Enzyme; ARB: Angiotensin II-Receptor Blockers; BMI: Body Max Index; CCB: Calcium-channel blockers; CI: Confident Interval; CVA: Cerebrovascular Accident; NSAIDs: Non-steroidal Anti-inflammatory Drugs; OR: Odds ratio; PAD: Peripheral Artery Disease; PPI: Proton-pump inhibitors; SD: Standard Deviation; TIA: Transient Ischemic Accident.

[*] Recorded as such or when patients were using nitrates.

[†] Recorded as such or when patients were using glucose-lowering drugs.

[§] Recorded as such or when patients were using lipid-lowering drugs.

[‖] Adjusted only for matching factors (age, sex, and calendar year).

## 3.1. SYSADOAs (CS and glucosamine) use and AMI risk

161 cases (0.68%) and 1,161 controls (0.99%) were current users of CS or glucosamine, which leads to an unadjusted OR of 0.68 (95%CI: 0.58–0.80), which hardly changed after full adjustment: AOR 0.69 (95%CI: 0.58–0.81). Such decreased risk disappeared upon discontinuation (recent and past users) (Table 2).

**Table 2. Risk of AMI associated with the use of SYSADOA.**

| | Cases (%) N = 23585 | Controls (%) N = 117405 | Non-adjusted OR[†] (95% CI) | Adjusted OR[§] (95% CI) |
|---|---|---|---|---|
| **SYSADOAs (all)** | | | | |
| Non users | 22606 (95.85) | 112202 (95.57) | 1 (Ref.) | 1 (Ref.) |
| Current | 161 (0.68) | 1161 (0.99) | 0.68 (0.58–0.80) | 0.69 (0.58–0.81) |
| Recent | 275 (1.17) | 1315 (1.12) | 1.03 (0.91–1.18) | 1.00 (0.87–1.14) |
| Past | 543 (2.30) | 2727 (2.32) | 0.99 (0.90–1.09) | 0.92 (0.84–1.02) |
| **Glucosamine** | | | | |
| Non users | 23035 (97.67) | 114584 (97.60) | 1 (Ref.) | 1 (Ref.) |
| Current | 79 (0.33) | 482 (0.41) | 0.81 (0.64–1.03) | 0.85 (0.66–1.08) |
| Recent | 129 (0.55) | 616 (0.52) | 1.04 (0.86–1.26) | 0.99 (0.81–1.20) |
| Past | 342 (1.45) | 1723 (1.47) | 0.99 (0.88–1.11) | 0.93 (0.83–1.06) |
| **Chondroitin sulfate** | | | | |
| Non users | 23027 (97.63) | 114323 (97.37) | 1 (Ref.) | 1 (Ref.) |
| Current | 89 (0.38) | 757 (0.64) | 0.57 (0.46–0.71) | 0.57 (0.46–0.72) |
| Recent | 172 (0.73) | 817 (0.70) | 1.04 (0.88–1.22) | 1.02 (0.86–1.21) |
| Past | 297 (1.26) | 1508 (1.28) | 0.98 (0.86–1.11) | 0.91 (0.80–1.04) |
| **Chondroitin sulfate + Glucosamine[*]** | | | | |
| Non users | 23456 (99.45) | 116705 (99.40) | 1 (Ref.) | 1 (Ref.) |
| Current | 7 (0.03) | 76 (0.06) | 0.46 (0.21–0.99) | 0.49 (0.22–1.08) |
| Recent | 47 (0.20) | 214 (0.18) | 1.10 (0.80–1.51) | 1.08 (0.77–1.50) |
| Past | 75 (0.32) | 410 (0.35) | 0.91 (0.71–1.17) | 0.86 (0.67–1.11) |

Abbreviations: CI: Confident Interval; OR: Odds ratio.

[*] Fixed-dose combination or concomitant use as separate drugs.

[†] Adjusted only for matching factors (age, sex, and calendar year).

[§] Adjusted for the matching factors (age, sex, and calendar year) plus the covariates shown in Table 1.

When CS and glucosamine were analysed separately, we observed that the association with a reduced risk was driven by CS (AOR 0.57; 95%CI: 0.46–0.72), while no risk reduction was observed with glucosamine (AOR 0.85; 95%CI: 0.66–1.08) (Table 2). In addition, when current use of CS was directly compared with current use of glucosamine, the protective effect of CS remained (AOR 0.64; 95%CI: 0.45–0.91) (S1 Table in S1 File). Due to the very small number of patients currently using combined CS and glucosamine (7 cases and 76 controls), the result associated with the combination was unprecise and non-significant, although a reduced risk was still suggested (AOR 0.49; 95%CI: 0.22–1.08).

## 3.2. SYSADOAs (CS and glucosamine) use and AMI risk: Duration of treatment and daily dose

Current use of SYSADOAs in accordance to treatment duration (less than 365 days, 365 days or more) was lower among cases (0.49% and 0.20%, respectively) than among controls (0.72% and 0.27%, respectively), leading to unadjusted OR of 0.67 (95%CI: 0.55–0.81) and 0.71 (95% CI: 0.52–0.97), respectively, as compared to non-users. After full adjustment, results were barely modified: 0.68 (95%CI: 0.56–0.84) and 0.69 (95%CI: 0.50–0.95), respectively. Table 3 shows the results by pharmacological group and active ingredient. As in the main analysis, the reduced risk was only observed for CS. No differences were observed by CS daily dose, though few patients used the lower dose (Table 4).

## 3.3. SYSADOA (CS and glucosamine) use and AMI risk in different subgroups

No evidence of statistical interaction with age, sex and concomitant use of NSAIDS was found (Fig 2 and S2 Table in S1 File). As for the CV risk profile, the reduced risk of AMI associated with CS was observed in both intermediate (AOR = 0.65; 95%CI: 0.46–0.91) and high (AOR = 0.61; 95%CI: 0.41–0.91) CV risk groups, while it was not observed in the low CV risk group (AOR = 1.11; 95%CI: 0.48–2.56). Specifically, among patients with antecedents of

**Table 3. Risk of AMI associated with the current use of chondroitin sulfate and glucosamine as compared to non-use according to duration of treatment.**

| | Cases (%) N = 23585 | Controls (%) N = 117405 | Non-adjusted OR* (95% CI) | Adjusted OR† (95% CI) |
|---|---|---|---|---|
| **SYSADOA (all)** | | | | |
| < 365 days | 115 (0.49) | 847 (0.72) | 0.67 (0.55–0.81) | 0.68 (0.56–0.84) |
| <91 days | 62 (0.26) | 476 (0.41) | 0.64 (0.49–0.84) | 0.67 (0.51–0.88) |
| 91–364 days | 53 (0.22) | 371 (0.32) | 0.70 (0.53–0.94) | 0.70 (0.52–0.95) |
| 365+ days | 46 (0.20) | 314 (0.27) | 0.71 (0.52–0.97) | 0.69 (0.50–0.95) |
| **Glucosamine** | | | | |
| < 365 days | 51 (0.22) | 356 (0.30) | 0.71 (0.53–0.96) | 0.76 (0.56–1.02) |
| <91 days | 28 (0.12) | 192 (0.16) | 0.73 (0.49–1.08) | 0.82 (0.55–1.23) |
| 91–364 days | 23 (0.10) | 164 (0.14) | 0.70 (0.45–1.08) | 0.69 (0.44–1.08) |
| 365+ days | 28 (0.12) | 126 (0.11) | 1.10 (0.73–1.66) | 1.09 (0.71–1.67) |
| **Chondroitin sulfate** | | | | |
| < 365 days | 66 (0.28) | 565 (0.48) | 0.57 (0.44–0.74) | 0.58 (0.45–0.75) |
| <91 days | 37 (0.16) | 332 (0.28) | 0.55 (0.39–0.77) | 0.54 (0.38–0.77) |
| 91–364 days | 29 (0.12) | 233 (0.20) | 0.61 (0.41–0.89) | 0.64 (0.43–0.95) |
| 365+ days | 23 (0.10) | 192 (0.16) | 0.57 (0.37–0.88) | 0.56 (0.36–0.87) |

Abbreviations: CI: Confidence Interval; OR: Odds ratio.

* Adjusted only for matching factors (age, sex, and calendar year).

† Adjusted for the matching factors (age, sex, and calendar year) plus the covariates shown in Table 1.

**Table 4. Risk of AMI associated with the current use of chondroitin sulfate as compared to non-use according to daily dose.**

| | Cases (%) N = 23585 | Controls (%) N = 117405 | Non-adjusted OR* (95% CI) | Adjusted OR† (95% CI) |
|---|---|---|---|---|
| **Chondroitin sulfate** | | | | |
| < 800 mg/24h | 8 (0.03) | 60 (0.05) | 0.64 (0.30–1.33) | 0.59 (0.28–1.27) |
| 800+ mg/24h | 63 (0.27) | 538 (0.46) | 0.57 (0.44–0.74) | 0.59 (0.45–0.77) |
| Unknown | 18 (0.08) | 159 (0.14) | 0.56 (0.34–0.92) | 0.52 (0.32–0.86) |

Abbreviations: CI: Confidence Interval; OR: Odds ratio.

* Adjusted only for matching factors (age, sex, and calendar year).

† Adjusted for the matching factors (age, sex, and calendar year) plus the covariates shown in Table 1.

angina pectoris we found an AOR of 0.06 (95%CI:0.01–0.48), though these data should be interpreted with caution due to the low numbers of patients (S3 Table in S1 File).

### 3.4. Sensitivity analysis

The inclusion of prevalent users leads to a lesser or similar risk reduction of AMI among current users of CS and a significant reduction in glucosamine current users (S4 Table in S1 File).

All variables included in our model had a VIF below 1.7 and the average VIF was 1.18, ruling out collinearity (S5 Table in S1 File).

## 4. Discussion

In this large population-based case-control study, carried out in Spain, within a primary cohort of 3,764,470 patients, from which more than 23,000 incident AMI cases emerged, we

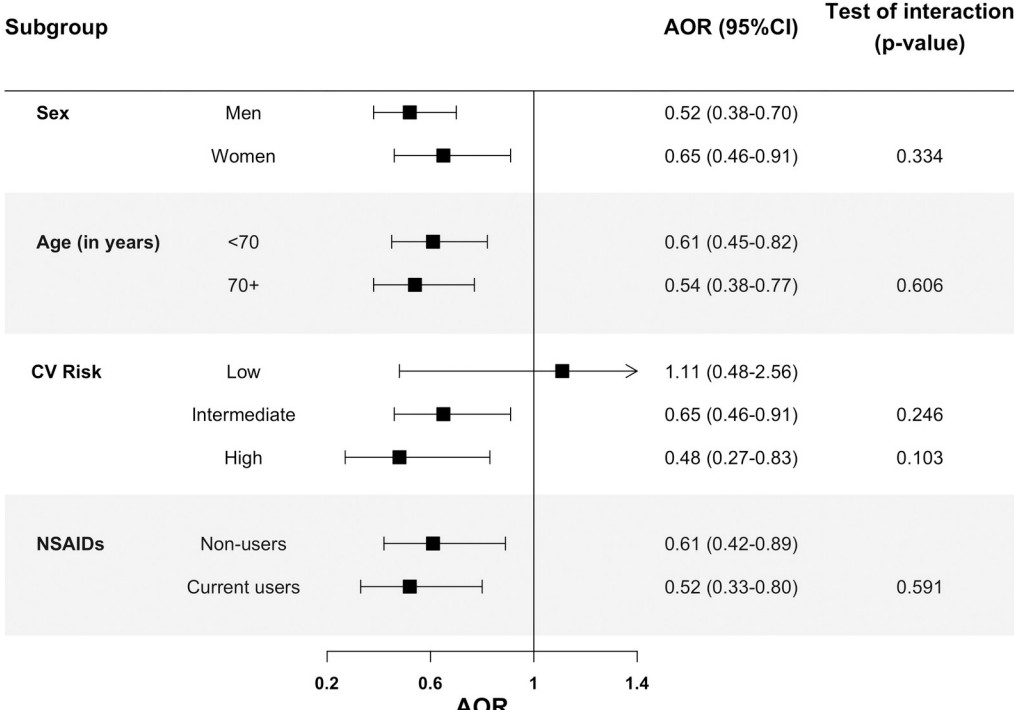

**Fig 2. Risk of acute myocardial infarction associated to current use of chondroitin sulfate by sex, age, background Cardiovascular (CV) risk and concomitant use of NSAIDs.** Abbreviations: AOR: Adjusted Odds Ratio, CI: Confidence Interval; CV: Cardiovascular; NSAIDs: Non-steroidal Anti-inflammatory Drugs.

obtained the following main findings: (1) current use of CS is associated with a reduced risk of AMI (40% reduction) which suggests a substantial cardioprotection effect; (2) such effect is observed in both men and women, irrespective of age and in both short-term and long-term users; (3) likewise, it is observed in subjects with intermediate and high CV risk, but not in patients at low-risk; and (4) it is observed in patients who are non-users of NSAIDs but also in those who are concomitantly using NSAIDs; 5) current use of glucosamine does not increase or decrease the risk of AMI.

An important result from this study is that the protective effect associated with CS was not observed for glucosamine, both in the main analysis and in all analyses performed on different sub-groups; also, the direct comparison CS vs. glucosamine suggested a strong AMI risk reduction among CS current users. Such direct comparison is an ideal situation as patients in both groups are highly comparable minimizing the possibility of bias. Another remarkable fact is that the observed protective effect of CS is not due to an NSAIDS sparing effect as it remains in both NSAIDs users and non-users.

The different results found with CS and glucosamine are interesting and suggest that these drugs have different actions on the CV system. Both are natural compounds but chemically different: glucosamine is an amino sugar and chondroitin sulfate is a glycosaminoglycan (GAG) involved in proteoglycan (PGs) biosynthesis [17,18]. CS, in contrast with glucosamine, is present in the extracellular matrix, particularly in cartilage, skin, blood vessels, ligaments and tendons. There is increasing evidence of the role played by PGs and GAGs in atherosclerosis [38]. The luminal surface of the endothelium is covered with a gel-like layer, glycocalyx, made up of glycoproteins (GAGs and PGs), CS being one of the most abundant GAGs [38]. The importance of glycocalyx lies in the fact that it takes part in multiple physiological processes of the endothelium: filtration of fluid and macromolecules, vascular tone regulation and hemostasis, as well as regulation of neutrophil migration across the endothelium [38]. For all these reasons, in recent years, numerous studies have hypothesized that the glycocalyx PGs and GAGs may play a role in atherosclerosis onset and progression [38,39]. Excess ROS (reactive oxygen species) in diabetes mellitus, hypertension and atherosclerosis triggers some mechanisms of pathogenicity that translate into endothelial dysfunction [38]. ROS have a negative direct effect on glycocalyx and GAGs causing depolymerization and shedding [38]. On the other hand, there is increasing evidence that GAGs and CS administration has a positive effect on vascular diseases and endothelial dysfunction: they are able to rebuild glycocalyx, have anti-inflammatory and anti-apoptotic effects, are heparinase and metalloprotease inhibitors, and have a protective effect on glucose-induced damage [38,40]. As for animal models of atherosclerosis, Melgar-Lesmes *et al* [41] observed that in mice receiving CS, this component directly latches onto the atheromatous plaque, drastically reducing its size, recedes TNF effects, heals endothelial injury, and decreases the monocyte/macrophage differentiation into foam cells. Furthermore, mice receiving CS showed 100% survival in comparison to 85% control survival [41]. CS treatment also showed positive effects in other animal models of atherosclerosis [40]. The fact that the reduced risk of AMI was observed early in users of CS supports the idea that the cardioprotective effect observed may be related with a stabilization of the atherosclerotic plaque, that may be related with its anti-inflammatory effects [42,43]. Laboratory studies suggest that CS inhibits the transcription factor nuclear factor kappa B (NFkB) from translocating to the nucleus [40,42], a key factor in many inflammatory processes, including atherosclerosis. Also, it has been reported that CS inhibits inflammatory factors downstream of NFkB signaling, including IL-1β, IL-6, TNF-α, and PGE2, as well as COX-2 expression [42,44–46]. In a recent NHANES study of nearly 10,000 adults aged 25 and older, the authors observed that glucosamine use and CS use were each associated with significantly reduced levels of C-

reactive protein (CRP), an important biomarker of inflammation, with larger reductions in women [23].

A noteworthy fact is that CS and glucosamine are prescribed medication for OA in most of the European countries. However, in others, such as United States and Australia, are considered dietary supplements [47], frequently taken together on a daily basis [48]. In Spain, glucosamine and CS are medically prescribed and their use in combination is infrequent (as shown in the present study). Such circumstance makes the present study unique to study the differential effects of these two drugs.

The first evidence on the possible protective role played by CS in CV diseases in humans was raised by Morrison [30,31] in the 70s, in an open-label clinical trial carried out with 120 patients with ischemic heart disease, assigned in a 1:1 ratio to the experimental (CS) and control group. While 42 patients (70%) in the control group had one cardiac event per month and 14 (23%) died after a 6-year follow-up, only 6 (10%) patients receiving CS experienced an acute cardiac event, and only 4 (6.6%) died [30,31]. Nevertheless, no additional clinical trial using current quality standards has been carried out since then. In a previous and smaller case-control study carried out by our research group, we observed that patients chronically receiving CS showed a lower risk of AMI as compared to non-use [20].

The strengths of the present study are the following: (1) although the study is retrospective, the BIFAP database, in which this study was performed, prospectively collects patient data from primary health care, including patient history, and a complete record of prescriptions filled; (2) the sample size of the study was large and allowed us to estimate risks with reasonable precision; (3) researchers who conducted the validation of cases were fully blinded to drug exposure in order to avoid a differential misclassification of the event; (4) controls were randomly extracted from the underlying cohort to make sure they represent the population exposure, this way, avoiding a selection bias; and (5) only "new users" were considered thus avoiding a "prevalent user" bias [29].

Limitations of the study are the following: (1) despite our efforts to control for confounding factors, a residual confounding due to unknown or unmeasured factors may be present due to the observational nature of the study; (2) it is possible that fatal cases (before hospital or in-hospital) are underestimated in primary care records, but it is unlikely that a potential misclassification of death associated with the event is differential with respect to the exposure of interest, as all data are recorded by the PCP in a prospective manner; as a non-differential misclassification of the event is known distort the measure of association towards the null, [49] such potential error would not explain the results found; (3) exposure misclassification is unlikely because all prescriptions are filled through the computer and then completely recorded, but treatment adherence by patients cannot be assured; (4) it was not possible to carry out a robust analysis with the combination of glucosamine and CS as the exposure was very low; (5) regular use of CS can be a marker of a healthy lifestyle, and then the cardioprotection observed could be partly explained by a healthy-user effect; however, two facts are against this possibility as the main explanation for the cardioprotective effect observed: we did not find a similar effect with glucosamine (used by a comparable population regarding CV risk factors, see S6 Table in S1 File and S1 Fig in S1 File); and the risk reduction with CS was observed, precisely, in patients with history of CV diseases (including diabetes) and patients with CV risk factors, but not in those with low CV risk.

The results of the present study support a cardioprotective effect of chondroitin sulfate which was observed in both short-term and long-term users, in both men and women, in individuals over and under 70 years of age, and in patients at intermediate and high CV risk, while no protection is found in individuals at low CV risk. By contrast, no such effect is observed with glucosamine.

## Supporting information

**S1 File. Supplementary information on methods and data.**
(DOCX)

## Acknowledgments

The authors would like to thank the excellent collaboration of primary care practitioners participating in BIFAP. We are also in debt with the staff members of the BIFAP Unit. The database BIFAP is managed by the Spanish Agency for Medicines and Medical Devices and make the data available for free to professionals from Academia and the National Health System. The results, discussion and conclusions are from the authors and do not necessarily represent the position of their Institutions or the Spanish Agency for Medicines and Medical Devices.

## Author Contributions

**Conceptualization:** Ramón Mazzucchelli, Sara Rodríguez-Martín, Alberto García-Vadillo, Francisco J. de Abajo.

**Data curation:** Sara Rodríguez-Martín, Miguel Gil.

**Formal analysis:** Sara Rodríguez-Martín, Francisco J. de Abajo.

**Funding acquisition:** Ramón Mazzucchelli, Francisco J. de Abajo.

**Investigation:** Ramón Mazzucchelli, Sara Rodríguez-Martín, Alberto García-Vadillo, Miguel Gil, Antonio Rodríguez-Miguel, Diana Barreira-Hernández, Alberto García-Lledó, Francisco J. de Abajo.

**Methodology:** Sara Rodríguez-Martín, Miguel Gil, Antonio Rodríguez-Miguel, Francisco J. de Abajo.

**Project administration:** Sara Rodríguez-Martín, Francisco J. de Abajo.

**Resources:** Ramón Mazzucchelli, Sara Rodríguez-Martín, Francisco J. de Abajo.

**Software:** Sara Rodríguez-Martín.

**Supervision:** Ramón Mazzucchelli, Francisco J. de Abajo.

**Validation:** Sara Rodríguez-Martín, Miguel Gil, Antonio Rodríguez-Miguel, Diana Barreira-Hernández, Alberto García-Lledó, Francisco J. de Abajo.

**Writing – original draft:** Ramón Mazzucchelli, Sara Rodríguez-Martín, Francisco J. de Abajo.

**Writing – review & editing:** Ramón Mazzucchelli, Sara Rodríguez-Martín, Alberto García-Vadillo, Miguel Gil, Antonio Rodríguez-Miguel, Diana Barreira-Hernández, Alberto García-Lledó, Francisco J. de Abajo.

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
