## [Decision Letter · Decision Letter 0]

24 May 2021

PONE-D-21-07801

RISK OF ACUTE MYOCARDIAL INFARCTION AMONG NEW USERS OF CHONDROITIN SULFATE: A NESTED CASE-CONTROL STUDY

PLOS ONE

Dear Dr. Francisco Jose Abajo,

Thank you for submitting your manuscript to PLOS ONE. After careful consideration, we feel that it has merit but does not fully meet PLOS ONE’s publication criteria as it currently stands. Therefore, we invite you to submit a revised version of the manuscript that addresses the points raised during the review process.

We look forward to receiving your revised manuscript.

Kind regards,

Ping-Hsun Wu, M.D. PhD.

Academic Editor

PLOS ONE

Additional Editor Comments:

The validation of acute myocardial infarction could be described in the article. Besides, the explanation of chondroitin sulfate could be discussed more in the article.

Journal Requirements:

[FdA has received unrestricted research grants from Instituto de Salud Carlos III, Instituto Teófilo Hernando, Chiesi and Sanofi-Pasteur for other projects in the last 5 years. The rest of authors have no conflicts of interest to declare. The results, discussion, and conclusions are from the authors and do not necessarily represent the position of the Spanish Agency for Medicines and Medical Devices.].

Reviewers' comments:

Reviewer's Responses to Questions

**Comments to the Author**

1. Is the manuscript technically sound, and do the data support the conclusions?

Reviewer #1: Yes

Reviewer #2: Yes

2. Has the statistical analysis been performed appropriately and rigorously? 

Reviewer #1: I Don't Know

Reviewer #2: Yes

3. Have the authors made all data underlying the findings in their manuscript fully available?

Reviewer #1: Yes

Reviewer #2: Yes

4. Is the manuscript presented in an intelligible fashion and written in standard English?

Reviewer #1: Yes

Reviewer #2: Yes

5. Review Comments to the Author

Reviewer #1: 1.There are so many covariates in your study. The issue of multicollinearity should be considered. For example, the angina pectoris may be due to hypertension or aortic stenosis (not included in your study) instead of coronary artery disease (not included in your study). Hence, it is better to evaluate the multicollinearity between covariates and showed the results in supplementary data.

2.Acute myocardial infarction is the most critical clinical spectrum of coronary artery disease. Patients with coronary artery disease may be prone to acute myocardial infarction. It is a pity that coronary artery disease is not included as a covariate in your study. Maybe you can perform another analysis to evaluate the protective effect of chondroitin sulfate, in terms of acute myocardial infarction, in patients with coronary artery disease.

Reviewer #2: The authors used a Spanish database (BIFAP) to explore whether chondroitin sulfate or glucosamine can reduce the incidence of AMI. The authors concluded that chondroitin sulfate could cut 46% risk of AMI, while glucosamine offer neutral cardiovascular protection. The favorable effects of chondroitin sulfate were observed in short- and long-term course, both sexes, and age > and < 70 year-old.

Major comments:

1. Since the BIFAP database is from primary care, how to validate the accuracy of diagnosis of AMI?

2. Is it possible that the patient did not return to primary care service after AMI, such as death? If so, please describe the impact on the analysis.

3. Chondroitin sulfate reduce the risk of AMI by 46%, which seemed better than the report of statin in primary prevention for major adverse cardiovascular events. What are the possible explanations? And use other clinical data from chondroitin sulfate as a reference should be considered.

Minor comments:

1. Do doctors have different considerations in choosing patients to use chondroitin sulfate or glucosamine? If so, is it likely to affect the analysis results?

6. PLOS authors have the option to publish the peer review history of their article (what does this mean?). If published, this will include your full peer review and any attached files.

Reviewer #1: No

Reviewer #2: No

---

## [Author Response · Author response to Decision Letter 0]

2 Jun 2021

Additional Editor Comments:

The validation of acute myocardial infarction could be described in the article. 

Replay:

Thanks. We have passed the text related to case validation from the supplementary material to the main text. Please note that new references have been included, with the consequent re-numbering of cites. The new text reads as follows:

1.1. Data Source and Study Design

We performed a case-control study nested in a primary cohort obtained from BIFAP (Base de datos para la Investigación Farmacoepidemiológica en Atención Primaria), a primary health care database from Spain (see Appendix A) [33] during the study period from January 1st, 2002, to December 31st, 2015. BIFAP contains anonymised electronic records on clinical events, prescriptions and laboratory tests, among others, that are recorded routinely by primary care practitioners. BIFAP-2016, which was used in this study, contains data from 7.6 million of patients (38.6 million subjects per year) with an average follow-up of 5.1 years, from nine different Spanish autonomous communities out of a total of 17. BIFAP reflects the distribution of the Spanish population by sex and age, and has been validated through numerous pharmacoepidemiological studies[20,34], therefore obtaining results that are comparable to other known European databases. The study period covers 14 years (from January 1st, 2002, to December 31st, 2015. In a first step, we constructed a primary cohort composed of all the patients registered in the database that Patients were selected if they were aged 40 to 99 years, had at least 1 year of follow-up with their primary care physician and did not have a history of cancer or AMI. For all patients included in the primary cohort (n=3,764,470), the first day they fulfilled all the inclusion criteria was considered as the “start date”. Since then, follow-up was carried out until the occurrence of any of the following events: incident AMI, turning 100 years old, cancer diagnosis, death, or end of study period.

1.2. Selection of Cases and Controls

Among patients in the primary cohort, incident AMI cases were initially searched in the database through codes and texts in diagnosis fields. and were validated via a manual revision of the electronic medical records (Appendix B). As for disease classification, eight autonomous communities used the International Classification of Primary Care, Second Edition (ICPC-2), and one autonomous community used the International Classification of Diseases, Ninth Revision, Clinical Modification (ICD-9-CM). The search was done to identify all potential AMI cases in the primary care cohort which were defined by code ICPC-2 K75 (acute myocardial infarction), code ICD-9-CM 410.9 (myocardial infarction) or related terms (free text) in diagnostic field. Next, potential identified cases were clustered into homogenous subgroups according to available information and a random sample was extracted for each subgroup, manually validating a total of 600 cases. The validation consisted of looking for additional information in the patient's medical record (such as comments in the free text associated with the diagnosis, records of hospitalization, interventions performed on the patient such as angioplasty or thrombolysis, biochemical test results, electrocardiogram results, ...) that would confirm that it was an incident case of AMI. Validation was performed independently by two of the researchers who were blind to any drug exposure and discrepancies were resolved by consensus of the whole research team. For each sample of validated cases in the homogeneous subgroups, the positive predictive value (PPV) was calculated, and when this was greater than 80%, the entire subgroup was considered valid cases. Overall, the PPV of the study with a PPV of was 87.2% (95% CI: 84.1-89.8%). The date when the first record of AMI took place was designated as “index date”. Five controls per case were randomly selected from the underlying cohort following a risk-set sampling in which controls were individually matched with cases on exact age, sex and index date.

Besides, the explanation of chondroitin sulfate could be discussed more in the article.

We expand the explanation given in Discussion as follows:

…… CS treatment also showed positive effects in other animal models of atherosclerosis [40]. The fact that the reduced risk of AMI was observed early in users of CS supports the idea that the cardioprotective effect observed may be related with a stabilization of the atherosclerotic plaque, that may be related with its anti-inflammatory effects [42,43]. The anti-inflammatory effect has also been described for glucosamine, inhibiting NF-kB and reducing inflammation markers, but the rest of mechanisms above described are not shared with glucosamine. Laboratory studies suggest that CS inhibits the transcription factor nuclear factor kappa B (NFkB) from translocating to the nucleus [40,42], a key factor in many inflammatory processes, including atherosclerosis. Also, it has been reported that CS inhibits inflammatory factors downstream of NFkB signaling, including IL-1β, IL-6, TNF-α, and PGE2, as well as COX-2 expression [42, 44-46]. In a recent NHANES study of nearly 10,000 adults aged 25 and older, the authors observed that glucosamine use and CS use were each associated with significantly reduced levels of C-reactive protein (CRP), an important biomarker of inflammation, with larger reductions in women [23].

Please note that new references have been included, with the consequent re-numbering of cites:

23. Bell GA, Kantor ED, Lampe JW, Shen DD, White E. Use of glucosamine and chondroitin in relation to mortality. Eur J Epidemiol 2012;27:593–603. doi:10.1007/s10654-012-9714-6

40. Herrero-Beaumont G, Marcos ME, Sánchez-Pernaute O, Granados R, Ortega L, Montell E, et al. Effect of chondroitin sulphate in a rabbit model of atherosclerosis aggravated by chronic arthritis: Chondroitin sulphate in atherosclerosis. Br J Pharmacol 2009;154:843–51. doi:10.1038/bjp.2008.113

42. Largo R, Martínez-Calatrava MJ, Sánchez-Pernaute O, Marcos ME, Moreno-Rubio J, Aparicio C, et al. Effect of a high dose of glucosamine on systemic and tissue inflammation in an experimental model of atherosclerosis aggravated by chronic arthritis. Am J Physiol Heart Circ Physiol 2009;297:H268-276. doi:10.1152/ajpheart.00142.2009

43. Navarro SL, White E, Kantor ED, Zhang Y, Rho J, Song X, et al. Randomized trial of glucosamine and chondroitin supplementation on inflammation and oxidative stress biomarkers and plasma proteomics profiles in healthy humans. PLoS One 2015;10:e0117534. doi:10.1371/journal.pone.0117534

44. Xu C-X, Jin H, Chung Y-S, Shin J-Y, Woo M-A, Lee K-H, et al. Chondroitin sulfate extracted from the Styela clava tunic suppresses TNF-alpha-induced expression of inflammatory factors, VCAM-1 and iNOS by blocking Akt/NF-kappaB signal in JB6 cells. Cancer Lett 2008;264:93–100. doi:10.1016/j.canlet.2008.01.022

45. Chou MM, Vergnolle N, McDougall JJ, Wallace JL, Marty S, Teskey V et al. Effects of chondroitin and glucosamine sulfate in a dietary bar formulation on inflammation, interleukin-1beta, matrix metalloprotease-9, and cartilage damage in arthritis. Exp Biol Med (Maywood) 2005;230:255–62. doi:10.1177/153537020523000405

46. Sakai S, Sugawara T, Kishi T, Yanagimoto K, Hirata T. Effect of glucosamine and related compounds on the degranulation of mast cells and ear swelling induced by dinitrofluorobenzene in mice. Life Sci 2010;86:337–43. doi:10.1016/j.lfs.2010.01.001

Reviewer #1:

1.There are so many covariates in your study. The issue of multicollinearity should be considered. For example, the angina pectoris may be due to hypertension or aortic stenosis (not included in your study) instead of coronary artery disease (not included in your study). Hence, it is better to evaluate the multicollinearity between covariates and showed the results in supplementary data.

Replay:

Modern statistical packages (such as STATA, which we have used to perform the statistical analyses), detect collinearity and when this happens the variables involved are automatically withdrawn. In none of the logistic regressions carried out in this work did the statistical program detect collinearity. Additionally, we have explored collinearity using the "collinearity diagnostics" of Stata which computes the parameter VIF (variance inflation factor). As a rule, when a variable has a VIF above 10 or the average VIF is greater than 6, suggests collinearity and then it is recommended to examine the causes and withdraw the involved variables from the model. All variables included in our model had a VIF below 1.7 and the average VIF was 1.18, ruling out collinearity.

As proposed by the reviewer, we have included this analysis in section “2.6. Statistical analysis”, in the final part of the results (lin. 267), as well as a table with the results in the supplementary material.

2.6. Statistical Analysis:

….. An analysis of potential collinearity was performed by measuring the variance inflation factor (VIF); according to this, collinearity is considered to be present when a variable has an independent VIF value above 10 or the mean VIF is above 6.

3. Results:

All variables included in our model had a VIF below 1.7 and the average VIF was 1.18, ruling out collinearity (S5 Table).

2.Acute myocardial infarction is the most critical clinical spectrum of coronary artery disease. Patients with coronary artery disease may be prone to acute myocardial infarction. It is a pity that coronary artery disease is not included as a covariate in your study. Maybe you can perform another analysis to evaluate the protective effect of chondroitin sulfate, in terms of acute myocardial infarction, in patients with coronary artery disease.

Thanks for this comment. We would like to clarify that in this study we excluded by design patients who had a previous myocardial infarction in order to assure that only first episodes were considered (incident cases), which is a cleaner approach than mixing new episodes with recurrences. Also, in the model we included the variable “angina pectoris” (see table 1) which was present in 2657 patients among cases (11.27%) and 5106 among controls (4.35%). This variable included any record of angina of any type, as well as the use of nitrates as a specific indicator of the disease. We recognize, as the reviewer remarks, that angina pectoris is a clinical diagnosis not always related to coronary artery disease (CAD), and may be other underlying causes (aortic stenosis, hypertension and others). However, these other causes of angina are much less frequent than CAD, so we considered that the variable “angina pectoris”(after excluding patients with a previous AMI) is a close representation of symptomatic CAD, bearing in mind that asymptomatic CAD is very frequent and nearly impossible to assess in a population-based study. Therefore, in our view, the variable “angina pectoris” widely overlaps with CAD (after excluding patients with a previous AMI), and then our results can be considered adjusted for CAD.

 In the second part of the comment, the reviewer propose to restrict the analysis to patients with CAD, which is an interesting proposal. In this point, it is important to note that we performed a stratified analysis by the “background cardiovascular risk”, in which the "high-risk" category includes patients with history of peripheral artery disease, angina pectoris, cerebrovascular accident or diabetes (see Methods, 2.6. Statistical analysis, for definitions and Figure 2 for results). In this stratum, we found an AOR of 0.48 (95%CI:0.27-0.83), still compatible with a protective effect even in this high-risk population. Notwithstanding, following the reviewer´s comment we have also analyzed the data specifically in patients with antecedents of angina and found an AOR of 0.06 (95%CI:0.01-0.48), again compatible with a high protective effect, though the low numbers precludes to draw firm conclusions. 

We propose to include this information in the main text as follows: 

3.3. SYSADOA (CS and glucosamine) use and AMI risk in different subgroups

No evidence of statistical interaction with age, sex and concomitant use of NSAIDS was found (Figure 2 and S2 Table). As for the CV risk profile, the reduced risk of AMI associated with CS was observed in both intermediate (AOR=0.65; 95%CI: 0.46-0.91) and high (AOR=0.61; 95%CI: 0.41-0.91) CV risk groups, while it was not observed in the low CV risk group (AOR=1.11; 95%CI: 0.48-2.56). Specifically, among patients with antecedents of angina pectoris we found an AOR of 0.06 (95%CI:0.01-0.48), though these data should be interpreted with caution due to the low numbers of patients (S3 Table).

The details are included in the supplementary material table S3. 

Reviewer #2: The authors used a Spanish database (BIFAP) to explore whether chondroitin sulfate or glucosamine can reduce the incidence of AMI. The authors concluded that chondroitin sulfate could cut 46% risk of AMI, while glucosamine offer neutral cardiovascular protection. The favorable effects of chondroitin sulfate were observed in short- and long-term course, both sexes, and age > and < 70 year-old.

Major comments:

1. Since the BIFAP database is from primary care, how to validate the accuracy of diagnosis of AMI?

Thank you for commenting on this crucial issue. In Spain, the National Health System (NHS) is universal, and the primary care physician (PCP) is the gatekeeper to the NHS, so any hospitalization or referral to specialists are routinely recorded by the PCP. Additionally, once the patient is discharged, he/she is normally followed-up by the PCP, who usually write the prescriptions set by the specialist in hospital. Further, we carried out an extensive validation work-up to ensure that we are picking up true incident AMI cases. This validation exercise, following Editor´s recommendations, is now described in detail in Methods, as follows:

1.1. Selection of Cases and Controls

Among patients in the primary cohort, incident AMI cases were initially searched through codes and texts in diagnosis fields. As for disease classification, eight autonomous communities used the International Classification of Primary Care, Second Edition (ICPC-2), and one autonomous community used the International Classification of Diseases, Ninth Revision, Clinical Modification (ICD-9-CM). The search was done to identify all potential AMI cases in the primary care cohort which were defined by code ICPC-2 K75 (acute myocardial infarction), code ICD-9-CM 410.9 (myocardial infarction) or related terms (free text) in diagnosis field. Next, potential identified cases were clustered into homogenous subgroups according to available information and a random sample was extracted for each subgroup, manually validating a total of 600 cases. The validation consisted of looking for additional information in the patient's medical record (such as comments in the free text associated with the diagnosis, records of hospitalization, interventions performed on the patient such as angioplasty or thrombolysis, biochemical test results, electrocardiogram results, ...) that would confirm that it was an incident case of AMI. Validation was performed independently by two of the researchers who were blind to any drug exposure and discrepancies were resolved by consensus of the whole research team. For each sample of validated cases in the homogeneous subgroups, the positive predictive value (PPV) was calculated, and when this was greater than 80%, the entire subgroup was considered valid cases. Finally, 24155 cases were included in the study with a PPV of 87.2% (95% CI: 84.1-89.8%). The date when the first record of AMI took place was designated as “index date”. Five controls per case were randomly selected from the underlying cohort following a risk-set sampling in which controls were individually matched with cases on exact age, sex and index date.

2. Is it possible that the patient did not return to primary care service after AMI, such as death? If so, please describe the impact on the analysis.

Thanks for this comment. If the patient survives, as we commented before, he/she should visit the PCP and, then, the case will be accurately recorded. When patient dies as a consequence of the AMI, either out of hospital (sudden death) or in hospital, it is possible that such event could be under-recorded. Although, deaths are administratively recorded, the cause of death is often not recorded or not recorded with the necessary specificity. Then, we should assume a certain degree of misclassification of fatal events among AMI cases. In our study, 1004 AMI cases were fatal (4.2%); bearing in mind that the in-hospital mortality in Spain ranged from 6.74% to 8.49% depending on the hospital type and care provided (Bertomeu et al, In-hospital Mortality Due to Acute Myocardial Infarction. Relevance of Type of Hospital and Care Provided. RECALCAR Study, Rev Esp Cardiol 2013, DOI: 10.1016/j.rec.2013.06.006), we consider that the under-recording should not be high. On the other hand, any misclassification of death associated with the event should be non-differential with respect to the exposure as the prescription and event data are recorded by the PCP in a prospective manner. As it is well-known, a non-differential misclassification of a binary event will normally bias the measure of association towards the null (Rothman, Greenland and Lash, 3rd ed, pp 142) and then it would not explain the results we found. To summarize this in the main text, we propose to add the following limitation (lin. 337): 

… (2) it is possible that fatal cases (before hospital or in-hospital) are underestimated in primary care records, but it is unlikely that a potential misclassification of death associated with the event is differential with respect to the exposure of interest, as all data are recorded by the PCP in a prospective manner; as a non-differential misclassification of the event is known distort the measure of association towards the null, [49] such potential error would not explain the results found. 

49. Rothman KJ, Greenland S, & Lash T. Modern Epidemiology, 3rd edn (Lippincott Williams and Wilkins, Philadelphia, PA, 2013). 

3. Chondroitin sulfate reduce the risk of AMI by 46%, which seemed better than the report of statin in primary prevention for major adverse cardiovascular events. What are the possible explanations? And use other clinical data from chondroitin sulfate as a reference should be considered.

The reviewer is perfectly right and we, honestly, did not expect such an impressive result. It is possible, however, that a healthy-user effect (e.g. CS users may have a healthier life-style), as we comment in the Discussion (page 14, line 344, new version), could partly contribute to the magnitude of the effect. Unfortunately, we are unable to adjust for a healthy lifestyle, as this variable is not properly measured or recorded in routine clinical practice (e.g. exercise, nutrition etc). Additionally, there may be other unknown or unmeasured confounding factors. To circumvent this limitation is why we carried out the head-to-head comparison of CS vs glucosamine, assuming that the clinical profile of CS and glucosamine users would be rather similar (see S6 table), likely including life-style factors. In such comparison we obtained an AOR of 0.64 (95%CI: 0.45-0.91), a weaker association than the one found when CS users were compared to non-users and probably much closer to the true effect of CS. As this question is properly discussed (Discussion sections, page 14, line 344, new version) we do not propose adding new comments. 

Minor comments:

1. Do doctors have different considerations in choosing patients to use chondroitin sulfate or glucosamine? If so, is it likely to affect the analysis results?

To the best of our knowledge, both drugs are indicated in Spain for the same indication (symptomatic treatment of osteoarthritis) and used for subjects with a similar clinical profile (as shown in supplementary Table S6 and supplementary figure S1 – new version) . So, “a priori” there are no specific patterns that may impact on the risk of acute myocardial infarction.

---

## [Decision Letter · Decision Letter 1]

16 Jun 2021

RISK OF ACUTE MYOCARDIAL INFARCTION AMONG NEW USERS OF CHONDROITIN SULFATE: A NESTED CASE-CONTROL STUDY

PONE-D-21-07801R1

Dear Dr. Francisco Jose Abajo,

We’re pleased to inform you that your manuscript has been judged scientifically suitable for publication and will be formally accepted for publication once it meets all outstanding technical requirements.

Kind regards,

Ping-Hsun Wu, M.D. PhD.

Academic Editor

PLOS ONE

Additional Editor Comments (optional):

All suggestions had been revised accordingly. This paper is available for publication.

Reviewers' comments:

Reviewer's Responses to Questions

**Comments to the Author**

1. If the authors have adequately addressed your comments raised in a previous round of review and you feel that this manuscript is now acceptable for publication, you may indicate that here to bypass the “Comments to the Author” section, enter your conflict of interest statement in the “Confidential to Editor” section, and submit your "Accept" recommendation.

Reviewer #2: All comments have been addressed

2. Is the manuscript technically sound, and do the data support the conclusions?

Reviewer #2: Yes

3. Has the statistical analysis been performed appropriately and rigorously? 

Reviewer #2: Yes

4. Have the authors made all data underlying the findings in their manuscript fully available?

Reviewer #2: Yes

5. Is the manuscript presented in an intelligible fashion and written in standard English?

Reviewer #2: Yes

6. Review Comments to the Author

Reviewer #2: The authors made a nice and comprehensive reply to my previous comments. I have no further comments. Thanks.

7. PLOS authors have the option to publish the peer review history of their article (what does this mean?). If published, this will include your full peer review and any attached files.

Reviewer #2: **Yes: **Shih-Sheng Chang M.D., Ph.D.

---

## [Editor Report · Acceptance letter]

2 Jul 2021

PONE-D-21-07801R1 

Risk of Acute Myocardial infarction among new users of Chondroitin Sulfate: A nested Case-Control study 

Dear Dr. de Abajo:

I'm pleased to inform you that your manuscript has been deemed suitable for publication in PLOS ONE. Congratulations! Your manuscript is now with our production department. 

Kind regards, 

on behalf of

Dr. Ping-Hsun Wu 

Academic Editor

PLOS ONE